# Every Step Counts: Decoding Trajectories as Authorship Fingerprints of dLLMs

Qi Li [1]  Runpeng Yu [1]  Haiquan Lu [1]  Xinchao Wang [1]

## Abstract

Discrete Diffusion Large Language Models (dLLMs) have recently emerged as a promising non-autoregressive paradigm, offering faster inference while achieving strong performance in code generation and mathematical reasoning tasks. In this work, we show that dLLMs' decoding mechanism not only improves utility but also enables effective model attribution: by analyzing a response's decoding trajectory, we can identify its source model and help mitigate risks from model misuse. A key challenge is the diversity of attribution scenarios, ranging from distinguishing different models to identifying different checkpoints or backups of the same model. To ensure broad applicability, we focus on two core questions: what information to extract from the decoding trajectory, and how to use it effectively. We first observe that per-step model confidence is ineffective, as the bidirectional nature of dLLMs causes mutual influence among decoded tokens, leading to highly redundant confidence signals that obscure structural information about decoding order and dependencies. To overcome this, we propose a novel information extraction scheme called the *Directed Decoding Map (DDM)*, which captures structural relationships between decoding steps and reveals model-specific behaviors. Furthermore, to fully leverage the extracted structure, we propose *Gaussian-Trajectory Attribution (GTA)*, which fits a cell-wise Gaussian distribution at each decoding position for each model and uses log-likelihood differences between trajectories as the attribution score. Extensive experiments across diverse models, datasets and different model access assumptions validate the effectiveness of our approach. Code is available here.

## 1. Introduction

Authorship attribution has long been an important problem in natural language processing, with wide applications in criminal investigations (Chaski, 2005; Koppel et al., 2008; Grant, 2020), tracking terrorist threat (Cafiero & Camps, 2023; Budryk, 2019), and social media protection (Hazell, 2023; Barbon Jr et al., 2017; Sinnott & Wang, 2021). In the era of large language models (LLMs), the value of attribution becomes even more prominent, as it promotes responsibility assignment and thus supports efforts to combat misinformation and fraudulent reviews generated by LLMs (McGovern et al., 2024; Li et al., 2023; Antoun et al., 2023; Lin et al., 2024). Typically, this task is approached as a binary classification problem (Fagni et al., 2021; Jawahar et al., 2020; Mitchell et al., 2023; Lin et al., 2024). Prior studies have explored distinguishing human-written text from LLM-generated text (Ji et al., 2024; Clark et al., 2021), while others have focused on detecting and attributing outputs from different LLMs (Li et al., 2023; McGovern et al., 2024). The latter, also known as Model Attribution (MA) (Antoun et al., 2023), is considerably more challenging due to the similarities in model architectures and training corpus across LLMs.

In this work, we provide the first study of the MA problem in the context of a new class of large language models, namely discrete diffusion large language models (dLLMs) (Nie et al., 2025; Ye et al., 2025; Yu et al., 2025). Unlike autoregressive generation (OpenAI , 2024; Gemini Team, 2025; DeepSeek-AI, 2025), dLLMs treat generation as an iterative decoding process over discrete token sequences (Yu et al., 2025). At each decoding step, any position of the token sequence may be unmasked from a mask token into a concrete token. This decoding process naturally forms a trajectory and introduces dependencies across decoding steps. It also captures, in a finer-grained manner, the distinctive characteristics that different models exhibit when responding to the same text prompt. Importantly, emerging commercial diffusion-LLM APIs like Inception's Mercury API (Khanna et al., 2025) already expose intermediate decoding trajectories. We show that such unique properties of dLLM decoding trajectories offer a novel basis for addressing the model attribution problem.

Specifically, to fully exploit the fine-grained structural infor-

---

[1]National University of Singapore. Correspondence to: Xinchao Wang <xinchao@nus.edu.sg>.

*Proceedings of the 43rd International Conference on Machine Learning*, Seoul, South Korea. PMLR 306, 2026. Copyright 2026 by the author(s).

mation embedded in the decoding process, we identify two fundamental problems: *how to effectively extract information from the decoding trajectory, and how to utilize such information for model attribution.* To solve these problems, we firstly introduce the *Directed Decoding Map (DDM)* for information extraction. The core motivation behind DDM is that different models exhibit stable differences in how newly decoded tokens interact with previously decoded tokens during generation. By encoding whether a newly decoded token induces positive, negative, or mixed effects on the confidence of previously decoded tokens, as well as the direction of confidence changes for each decoded tokens, DDM transforms complex probabilistic dynamics into a structured representation. This representation reliably captures model-specific decoding characteristics and thereby provides a reliable basis for model attribution. Building on this, we propose a novel model attribution method named *Gaussian-Trajectory Attribution (GTA)*. In GTA, each model is queried with a local dataset to obtain its own collection of DDMs. Based on these DDMs, we fit cell-wise Gaussian distributions separately for each model. This procedure preserves the structural information encoded in DDMs and produces a compact probabilistic fingerprint of each model's decoding behavior. To attribute a target response, we compute the log-likelihood of its DDM under all the constructed model-specific distributions and assign it to the model with the highest likelihood.

We conduct extensive experiments under various settings, including distinguishing between different models as well as between different checkpoints or backups of the same model. We also investigate the influence of different token length and decoding strategies in dLLMs, along with ablation studies on both DDM and GTA. The results consistently demonstrate the strong capability of our method. We summarize our contributions as follows:

- We present the first exploration of model attribution for dLLMs and introduce the use of their distinctive decoding trajectories for lightweight yet reliable attribution. To this end, we design an information extraction scheme named *Directed Decoding Map (DDM)*, which reliably captures the structural and dependency information embedded in the decoding trajectory for a better attribution process.

- We propose a model attribution method named *Gaussian-Trajectory Attribution (GTA)*, which builds compact probabilistic fingerprints for each model via cell-wise Gaussian fitting and attributes a target response to the model with the highest likelihood.

- Extensive experiments across different dLLMs and various attribution settings clearly demonstrate the superiority of DDM and GTA. For example, even in the

highly restrictive case where two models are fine-tuned from the same checkpoint using identical configurations, the attribution AUC remains above 81%.

## 2. Related Works

**Discrete Diffusion Large Language Models.** Discrete Diffusion Large Language Models (dLLMs) (Nie et al., 2025; Ye et al., 2025; Yu et al., 2025) recently emerges as a promising paradigm for non-autoregressive (non-AR) language modeling. In tasks such as code generation (Deep-Mind, 2025; Inception Labs, 2025), planning (Ye et al., 2025), and Sudoku (Ye et al., 2025), dLLMs have been widely shown to achieve better performance than AR models. In contrast to AR generation, dLLMs treat generation as an iterative decoding process over discrete token sequences (Nie et al., 2025; Ye et al., 2025). This paradigm removes the left-to-right constraint, allowing parallel and structurally controllable generation with bidirectional attention. Some recent works also explore deeply into the scaling-up nature and decoding behavior of dLLMs (Yang et al., 2025; Cheng et al., 2025).

Due to the bidirectional nature of dLLMs, their iterative decoding process can be naturally viewed as a trajectory with strong structural information and contextual dependency. The use of historical information from the decoding process has also inspired some current efforts. For instance, DIJA (Wen et al., 2025) and PAD (Zhang et al., 2025) reformulates conventional jailbreak prompts into an interleaved mask-text format, compelling the model to generate unsafe outputs while maintaining contextual consistency. Another work (Xie et al., 2025) shows that dLLMs are more vulnerable to manipulation at the middle of the response than at the initial tokens and proposes MOSA, a reinforcement learning alignment strategy, which requires the model's generated middle tokens to align with a set of predefined safe tokens. In this work, we make the first attempt to use the history trajectory for model attribution. Rather than modifying it as in prior work, we regard the trajectory as a holistic signal and extract from it a representation that highlights model-specific information.

**Authorship Attribution.** Authorship attribution (Mitchell et al., 2023; Hu et al., 2023; Yang et al., 2024; Bao et al., 2024; Hans et al., 2024; Verma et al., 2024; Liang et al., 2022a;b; Wang et al., 2024; Zeng et al., 2024; Su et al., 2023) is a long-standing problem that was initially applied to distinguishing between human authors, with practical applications such as criminal investigations (Chaski, 2005; Koppel et al., 2008; Grant, 2020) and counterterrorism efforts (Cafiero & Camps, 2023; Budryk, 2019). With the rise of large language models (LLMs), and inspired by the Turing Test (Turing, 2007; Biever, 2023; Mei et al., 2024), research has expanded to distinguishing human-written from

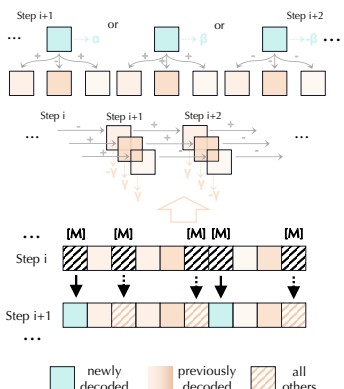

*Figure 1.* The DDM construction pipeline. [M] is the mask token.

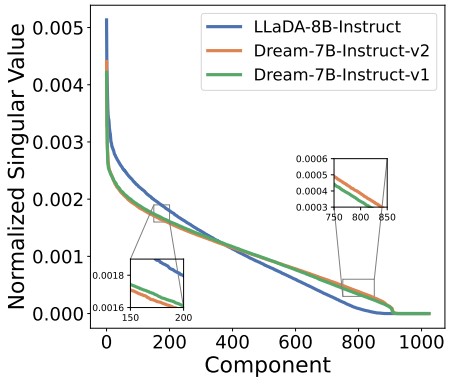

*Figure 2.* An SVD analysis on the structural information of DDMs.

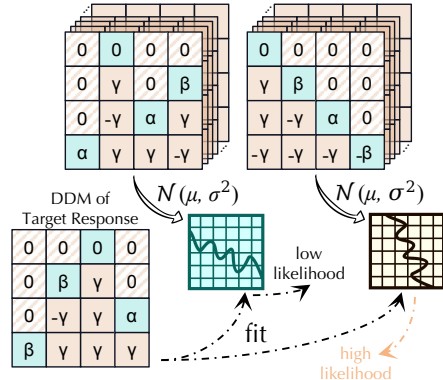

*Figure 3.* The GTA construction process and the attribution pipeline.

machine-generated text (Lin et al., 2024). Furthermore, model attribution (MA) further extends the concept of authorship attribution to the machine–machine setting (Li et al., 2023; Lin et al., 2024; Antoun et al., 2023), where the objective is to identify which specific model, or even which version of a model, was responsible for generating a given text. Existing model attribution methods can be broadly grouped into three categories: statistical feature-based methods (Sharma et al., 2018; Sari et al., 2017; Shrestha et al., 2017; Proisl et al., 2018), which rely on measures such as perplexity, n-grams, and entropy and are lightweight but often limited in performance; classifier-based methods (Solorio et al., 2011; Shao et al., 2019), which train discriminative models on texts from different sources and achieve higher accuracy but at the cost of efficiency and robustness to adversarial mimicking; and watermark-based methods (Kirchenbauer et al., 2023; Boenisch, 2021), which take a pre-hoc approach by embedding watermarks in the generation process and later verifying attribution by detecting these watermarks. In this work, we design a lightweight and reliable statistical feature-based method, for the first time leveraging the unique decoding trajectory of dLLMs to achieve effective model attribution.

## 3. Methods

In this section, we first introduce our proposed information extraction scheme. The core intuition is

that relying solely on first-order signals such as model confidence is insufficient to capture the structural information and cross-step dependencies embedded in the decoding trajectories of dLLMs. To address this, we propose a second-order representation, the *Directed Decoding Map (DDM)* in Section 3, which explicitly encodes the interdependencies among tokens and steps during decoding into a structured representation. Building on this, we further present our attribution method, *Gaussian-Trajectory Attribution (GTA)* in Section 3, which fully leverages the information in DDMs

by fitting cell-wise Gaussian distributions over the extracted trajectories, thereby obtaining a compact probabilistic fingerprint of each model's decoding behavior and enabling lightweight yet reliable model attribution.

**Directed Decoding Map Construction.** Consider a dLLM decoding process of $T$ steps producing a sequence of $L$ tokens. We define $c_i(j)$ as the confidence of position $j$ at step $i$ (if $j$ is not yet decoded at step $i$, then $c_i(j) = \varnothing$). The confidence change of a position in two consecutive steps is defined as $\Delta c(j) = c_{i+1}(j) - c_i(j)$.

Let $U_i = \{j \mid c_i(j) \neq \varnothing\}$ be the set of positions already decoded in step $i$, and $N_{i+1} = U_{i+1} \setminus U_i$ denotes the set of token positions newly decoded at step $i + 1$. We define the Directed Decoding Map entry at the $i$-th row, $j$-th position as $E_i(j)$. Each position falls into one of three categories: newly decoded positions $E_i(n)$, previously decoded positions $E_i(p)$, and all other positions $E_i(o)$. As a starting point, the values in the first row $E_1(j)$ are set to 0.

As illustrated in Figure 1, for each newly decoded position $n \in N_{i+1}$, the change it induces on the confidence of previously decoded positions $p \in U_i$ may exhibit three possible patterns: (i) all $\Delta c(p)$ are non-negative (confidence consistently increases), (ii) all $\Delta c(p)$ are non-positive (confidence consistently decreases), or (iii) a mixture of increases and decreases occurs. We define two distinct effect values $\alpha, \beta \in \mathbb{R}^+$, $\alpha \neq \beta$, and assign the effect value for token $n \in N_{i+1}$ as

$$E_{i+1}(n) = \begin{cases} \alpha, & \exists p, p' \in U_i : \Delta c(p) > 0, \ \Delta c(p') < 0, \\ \beta, & \forall p \in U_i : \Delta c(p) \geq 0, \\ -\beta, & \forall p \in U_i : \Delta c(p) \leq 0. \end{cases}$$

$$(1)$$

This design captures whether the effect of a new token is mixed, purely positive, or purely negative. For each previously decoded token $p$, once it has been decoded, subsequent steps may either reinforce its confidence or diminish

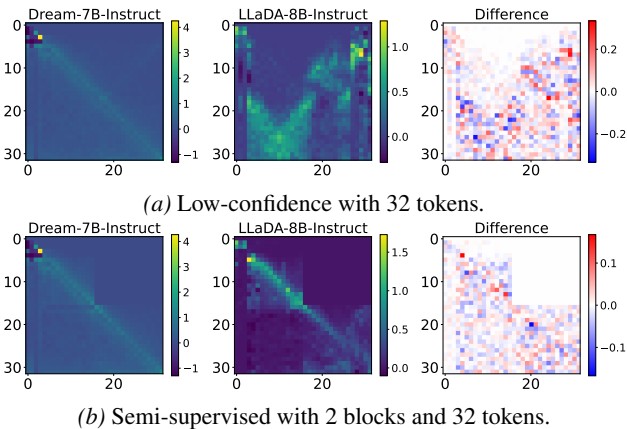

*(a)* Low-confidence with 32 tokens.

*(b)* Semi-supervised with 2 blocks and 32 tokens.

*Figure 4.* DDMs under two decoding strategies: *low-confidence decoding* and *semi-supervised decoding*. The models are instruction-tuned on GSM8K (Cobbe et al., 2021).

it. Hence, for $p \in U_i$, we assign

$$E_{i+1}(p) = \begin{cases} \gamma, & \Delta c(p) > 0, \\ -\gamma, & \Delta c(p) < 0, \end{cases} \quad (2)$$

where $\gamma \in \mathbf{R}^+, \gamma \neq \alpha \neq \beta$ is another effect value. This rule explicitly encodes whether the stability of an already decoded token is being strengthened or weakened at step $i + 1$. Finally, positions that remain masked carry no effect signal in this step. Thus, for any $o \notin U_{i+1}$, we set $E_i(o) = 0$. The final DDM is represented as a matrix $E \in \mathbb{R}^{T \times L}$, where each entry $E_i(j)$ reflects not only the decoding state but also the directionality of token-level influence across steps. Figure 4 shows DDMs for attribution between two models under two decoding strategies (Nie et al., 2025). For a fair comparison, both models (LLaDA-7B-Instruct (Nie et al., 2025) and Dream-8B-Instruct (Ye et al., 2025)) are instruction-tuned on GSM8K (Cobbe et al., 2021) with identical configurations, and the token number is set to 32. The effect values $\alpha, \beta, \gamma$ are set to 10, 0.5, 2, respectively (we report an ablation study on the effect values in Section 4, which shows that the actual effect value does not substantially affect the performance, a theoretical explanation of such invariance is given in Appendix J). As shown, even for different models under the same decoding strategy, DDMs can successfully capture structural differences between them, providing a reliable signal for attribution.

**Gaussian-Trajectory Attribution.** While the Directed Decoding Map (DDM) provides a structured representation of inter-step dependency, a central challenge remains: *how to effectively preserve and utilize these signals for attribution across models*. To better illustrate the importance of preserving the structural information in DDMs, we perform a SVD analysis (Golub & Van Loan, 2013) (more details are given in Appendix C). Specifically, DDMs

are flattened, concatenated into a matrix, mean-centered across features, and then decomposed via SVD. The resulting singular value spectrum characterizes how the variance of DDMs distributes across principal components. In Figure 2, the two versions of Dream-7B-Instruct are tuned from the same checkpoint under identical training configurations. As can be observed, different models exhibit almost identical spectra on the leading components, indicating that they share similar task-level and model-agnostic structures. However, clear discrepancies emerge in the middle and tail components, which correspond to low-energy yet highly discriminative directions. This observation highlights that *attribution signals are primarily encoded in the fine-grained structural patterns rather than in the dominant modes.* Consequently, applying dimensionality-reduction methods or highly noisy methods that retain only the leading components would inevitably discard these critical differences and significantly weaken attribution performance. To this end, we propose *Gaussian-Trajectory Attribution (GTA)*. By fitting Gaussian distributions to the cell-wise values of the DDM, GTA helps to obtain a compact probabilistic fingerprint that captures both local and global variation, enabling reliable model attribution.

Let a decoding trajectory be represented as a DDM matrix $E \in \mathbb{R}^{T \times L}$, where $T$ is the number of decoding steps and $L$ is the output token length. As illustrated in Figure 3, for each target model $M$, we query it with a local dataset to obtain $N$ trajectories $\{E^{(n)}\}_{n=1}^N$. These trajectories are then used to fit an independent Gaussian distribution for each cell $(t, l)$ by computing the empirical mean and variance across samples:

$$\mu_M(t, l) = \frac{1}{N} \sum_{n=1}^N E_{t,l}^{(n)},$$

$$\sigma_M^2(t, l) = \frac{1}{N} \sum_{n=1}^N \left( E_{t,l}^{(n)} - \mu_M(t, l) \right)^2. \quad (3)$$

After the cell-wise Gaussian construction, given a DDM of target response $E^* \in \mathbb{R}^{T \times L}$, its log-likelihood under model $M$ can be formalized as:

$$\ell_M(E^*) = -\frac{1}{2} \sum_{t=1}^T \sum_{j=1}^L \left[ \frac{\left( E_{t,l}^* - \mu_M(t, l) \right)^2}{\sigma_M(t, l)^2} \right.$$

$$\left. + \log \left( 2\pi \sigma_M(t, l)^2 \right) \right]. \quad (4)$$

For a candidate set of target models $\{M_1, \dots, M_K\}$, we compute the log-likelihood scores $\ell_{M_k}(E^*)$ for each $M_k$. The decision is then made by comparing likelihoods:

$$\hat{M}(E^*) = \arg \max_{M_k} \ell_{M_k}(E^*), \quad (5)$$

*Table 1.* Attribution Results under the three different setups. LLaDA-8B-Instruct (Nie et al., 2025) is used as the model to be attributed here in IRA and CCA. Semi-supervised decoding is used as the decoding strategy here, where the token length is set to be 32 and the block size is 16. Within each method, the best-performing information representation scheme is underlined, while the column-wise best results are highlighted in green .

| Scenario | Method | Information | GSM8K | | | | CodeAlpaca-20K | | | |
|---|---|---|---|---|---|---|---|---|---|---|
| | | | AUC | TPR@5%FPR | TPR@1%FPR | Acc. | AUC | TPR@5%FPR | TPR@1%FPR | Acc. |
| CMA | Perplexity | | 69.52 | 16.73 | 11.11 | 72.08 | 69.66 | 23.16 | 11.06 | 65.37 |
| | Clustering | confidence | 70.68 | 16.41 | 4.10 | 67.22 | 70.11 | 33.94 | 16.30 | 67.40 |
| | | filtered confidence | 69.68 | 21.01 | 3.52 | 64.83 | 63.33 | 12.75 | 2.07 | 61.58 |
| | | DDM | 97.34 | 94.38 | 90.37 | 95.47 | 97.83 | 92.88 | 71.53 | 94.12 |
| | Distance | confidence | 99.38 | 98.22 | 87.87 | 96.94 | 89.81 | 33.54 | 13.50 | 84.53 |
| | | filtered confidence | 99.85 | 99.91 | 98.66 | 98.91 | 89.59 | 42.73 | 8.44 | 82.60 |
| | | DDM | 99.92 | 99.96 | 98.93 | 99.04 | 97.47 | 84.98 | 64.19 | 92.35 |
| | GTA | confidence | 99.43 | 99.38 | 99.33 | 99.44 | 96.45 | 86.07 | 35.04 | 91.85 |
| | | filtered confidence | 99.66 | 99.02 | 95.67 | 97.99 | 95.66 | 84.22 | 65.11 | 90.35 |
| | | DDM | 99.95 (↑ 30.43) | 99.96 | 99.85 | 99.84 | 98.94 (↑ 35.61) | 98.22 | 92.03 | 97.15 |
| IRA | Perplexity | | 51.58 | 1.69 | 0.45 | 56.09 | 41.72 | 2.16 | 0.57 | 50.02 |
| | Clustering | confidence | 52.23 | 6.24 | 1.52 | 52.70 | 50.60 | 4.62 | 0.67 | 51.06 |
| | | filtered confidence | 60.22 | 7.23 | 1.92 | 59.10 | 51.84 | 12.59 | 6.64 | 51.59 |
| | | DDM | 61.15 | 7.00 | 1.38 | 59.48 | 53.82 | 6.71 | 2.16 | 52.93 |
| | Distance | confidence | 68.31 | 20.03 | 5.40 | 62.85 | 54.98 | 8.27 | 2.29 | 54.22 |
| | | filtered confidence | 68.91 | 22.84 | 7.00 | 64.27 | 55.18 | 6.27 | 1.07 | 54.64 |
| | | DDM | 79.49 | 33.05 | 19.18 | 72.26 | 58.73 | 10.72 | 1.24 | 57.01 |
| | GTA | confidence | 76.79 | 22.30 | 3.79 | 69.13 | 58.31 | 6.34 | 2.39 | 57.28 |
| | | filtered confidence | 80.35 | 31.76 | 5.17 | 73.26 | 54.98 | 8.27 | 2.29 | 54.22 |
| | | DDM | 81.75 (↑ 30.37) | 38.22 | 3.84 | 74.00 | 65.05 (↑ 23.33) | 14.35 | 3.72 | 60.88 |
| CCA | Perplexity | | 49.32 | 6.33 | 1.20 | 52.03 | 40.89 | 4.09 | 0.78 | 50.54 |
| | Clustering | confidence | 49.90 | 3.08 | 0.58 | 52.12 | 51.06 | 4.70 | 0.83 | 51.29 |
| | | filtered confidence | 51.03 | 6.47 | 1.61 | 52.05 | 51.03 | 4.57 | 0.70 | 51.14 |
| | | DDM | 56.33 | 7.63 | 1.56 | 54.57 | 53.79 | 7.51 | 1.94 | 53.49 |
| | Distance | confidence | 61.79 | 5.71 | 1.83 | 59.03 | 59.36 | 5.83 | 1.41 | 60.84 |
| | | filtered confidence | 57.34 | 6.74 | 1.56 | 56.47 | 54.68 | 9.53 | 2.39 | 54.46 |
| | | DDM | 65.84 | 8.61 | 1.43 | 61.24 | 59.47 | 11.15 | 3.05 | 61.14 |
| | GTA | confidence | 64.50 | 13.60 | 3.70 | 60.79 | 53.86 | 5.83 | 1.04 | 53.19 |
| | | filtered confidence | 59.53 | 8.83 | 1.87 | 58.14 | 54.13 | 6.03 | 1.02 | 53.36 |
| | | DDM | 66.91 (↑ 17.59) | 14.50 | 4.68 | 64.92 | 62.64 (↑ 21.75) | 6.38 | 1.31 | 61.78 |

i.e., we attribute the trajectory to the model under which it achieves the highest likelihood. Equivalently, when discriminating between two models $M_a$ and $M_b$, we define the attribution score as:

$$s(E^*; M_a, M_b) = \ell_{M_a}(E^*) - \ell_{M_b}(E^*), \quad (6)$$

where the sign of $s$ determines the attribution while the scale reflects the confidence of current attribution. GTA is lightweight yet powerful: (i) it preserves the structural information encoded by the DDM without collapsing it into coarse statistics, (ii) it produces compact probabilistic fingerprints that are easily comparable across models, and (iii) it enables fine-grained attribution even among models with similar architectures or training corpora.

## 4. Experiments

**Models and Datasets.** In our main experiments, we consider LLaDA-8B-Instruct (Nie et al., 2025) and Dream-7B-Instruct (Ye et al., 2025), along with multiple variants constructed under different attribution setups. For

dLLMs, direct attribution across distinct models is relatively straightforward due to their different decoding strategies. To establish a fairer and more challenging comparison, we instruction-tune both models on identical datasets with the same training configuration (details are given in Appendix D). For tuning and evaluation, we use GSM8K (Cobbe et al., 2021) for math reasoning and CodeAlpaca-20K (Chaudhary, 2023) for code generation, following the LLaDA training pipeline (Nie et al., 2025). Each dataset is split 7:3, with the larger part for training and the smaller for the local dataset. We adopt two decoding strategy: *Low-confidence* decoding and *Semi-supervised* decoding (Nie et al., 2025; Ye et al., 2025), and ensure that each model under attribution uses the same strategy. We use ⟨#tokens, block size⟩ to denote the combination of token length and block size. We consider five combinations: ⟨32, 32⟩ and ⟨64, 64⟩ for Low-confidence decoding, ⟨32, 16⟩, ⟨64, 32⟩, ⟨64, 16⟩ for Semi-supervised decoding.

**Attribution Scenario.** We consider three challenging yet important attribution scenarios: **(i)** across different models, referred to as Cross-Model Attribution (CMA); **(ii)** between

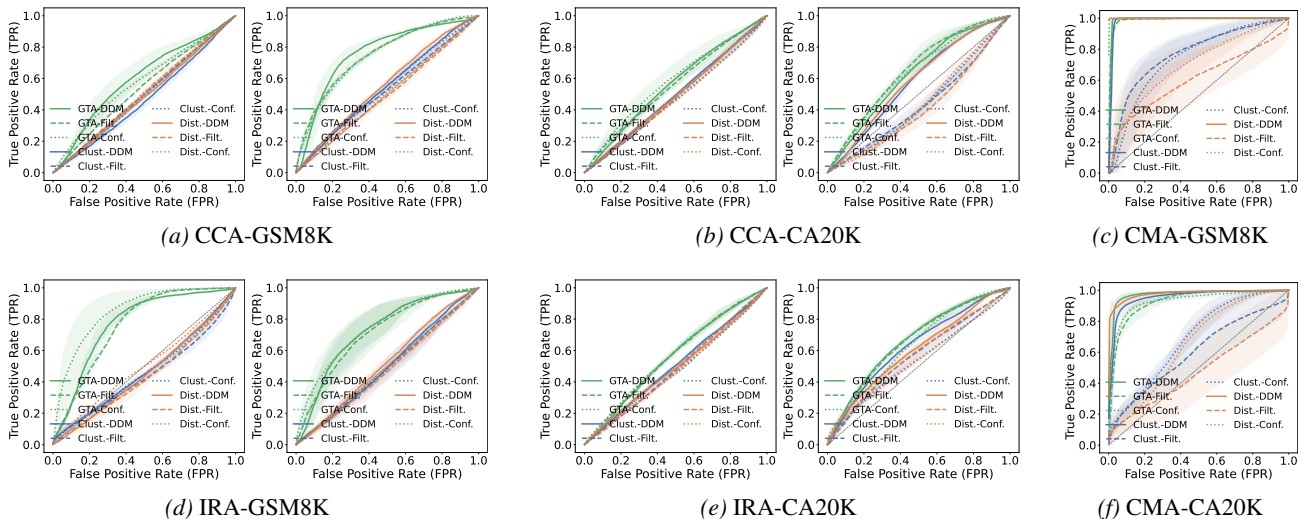

*Figure 5.* ROC curves of different attribution method–information extraction scheme combinations. *CA20K* abbreviates CodeAlpaca-20K. *Conf.*, *Filt.*, and *Clust.* stand for confidence, filtered confidence, and clustering. Attribution methods are distinguished by color, and extraction schemes by line style. Results are averaged over token lengths and decoding strategies. In subfigures (a)–(d), results on LLaDA are presented on the left, while those on Dream are on the right.

independent runs of the same model initialized from the same checkpoint and tuned using identical training configuration, referred to as Independent-Run Attribution (IRA); and **(iii)** across checkpoints of the same training trajectory at different epochs, referred to as Cross-Checkpoint Attribution (CCA). In our CCA setting, one model is fully trained for 20 epochs, while the other is the checkpoint saved halfway through training (at 10 epochs). Results under smaller intervals are given in Appendix H.

**Baselines.** We setup several baseline methods that can be fairly compared in our setting. For information extraction, *Confidence* directly uses the model's predicted probabilities over all positions at each decoding step, without distinguishing between decoded and masked tokens. *Filtered Confidence*, in contrast, leverages the decoded tokens and mask out unfinished positions, thereby retaining only the confidence scores of tokens that have been actually generated. For attribution method, *Clustering* applies unsupervised clustering (in our case, we use DBSCAN (Schubert et al., 2017) with Euclidean distance, an epsilon of 0.8, and a minimum of 20 points to form a cluster) over the trajectory features of responses from different models, and then uses cluster proximity to assign attributing labels. *Distance* computes the Euclidean distances between each target response and the average representations of each model, and uses the relative distance margins as the attribution score. *Perplexity (PPL)* (Alon & Kamfonas, 2023; Ankner et al., 2024) further aggregates the predicted token probabilities into a score by computing the exponential of the average negative log-likelihood over the response.

**Evaluation Metric.** We adopt AUC, TPR@Low% FPR, and Accuracy (Acc.) as evaluation metrics. AUC captures

overall attribution performance across thresholds, while TPR@Low% FPR emphasizes performance under strict false-positive control, which is crucial in high-cost misattribution scenarios. Accuracy provides an intuitive measure of overall correctness and complements the other metrics.

### 4.1. Main Results

We first provide a global overview of the performance trends across different combinations of attribution methods and information extraction schemes. The results are summarized in Table 1, where we follow prior work by formulating the problem as a binary classification task, i.e., attributing between two models. The token length is set to 32 and the block size to 16. Several key observations can be drawn from the results: **(i).** Among the three attribution scenarios, CMA proves to be the easiest, while CCA is the most challenging. This aligns with intuition: differences in decoding behavior across different models (CMA) are generally more pronounced than those between checkpoints or backups of the same model. Moreover, in CCA, the compared models can be regarded as one model being further fine-tuned from the other, whereas in IRA the compared models share the same initialization and undergo a fine-tuning process with same configuration. Consequently, the inter-model gap in CCA is usually smaller than that in IRA. **(ii).** Across different methods, DDM consistently demonstrates superior performance, yielding roughly a 10% AUC improvement over others under almost all settings. **(iii).** GTA achieves consistently stronger attribution performance than alternative methods, with the combination of GTA and DDM (i.e., our attribution method) delivering the best results.

In addition, to provide a more intuitive comparison, we

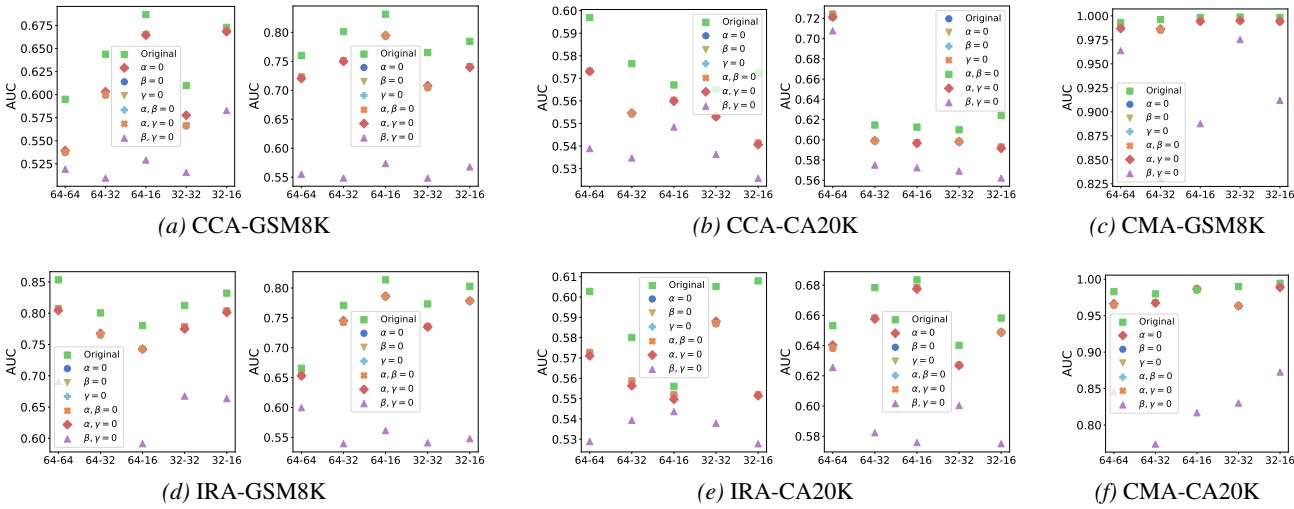

*Figure 6.* This experiment focuses on: (i) the impact of zeroing out specific effect values, and (ii) the effect of decoding strategy and token length on performance. In subfigures (a)–(d), results on LLaDA are presented on the left, while those on Dream are on the right.

report the ROC curves of different methods under various settings. Results are given in Figure 5, where *CA20K* denotes the abbreviation of CodeAlpaca-20K. *Conf.*, *Filt.*, and *Clust.* denote confidence, filtered confidence, and clustering, respectively. Each attribution method is represented by a distinct color, while each information extraction scheme is indicated by a specific line style. Results are averaged across the five considered token lengths and decoding strategies. Within subfigures (a), (b), (d) and (e), the left panel shows the results under LLaDA, while the right panel corresponds to Dream. The shaded regions indicate the variance across five token length and decoding strategy settings we consider (see Section 4), As can be observed, all three information extraction schemes under GTA outperform the baselines, and DDM consistently enables more stable attribution. Specifically, in the more challenging CCA and IRA settings, nearly all baselines perform close to random guessing. In contrast, for the relatively easier CMA setting, some baselines achieve moderate AUC but remain suboptimal; only a few methods combined with DDM, such as Dist.-DDM on GSM8K and Dist.-DDM / Clust.-DDM on CodeAlpaca-20K, attain strong performance.

### 4.2. Ablation Study and Analysis

**Structural Information of DDM.** We conduct an experiment to investigate the importance of the structural information extracted by DDM from decoding trajectories. Specifically, we zero out each of the three effect values individually and in pairs, resulting in six different variants. As shown in Figure 6, we evaluate these variants along with the original DDM (denoted as Original) under five combinations of token length and decoding strategy, with different colors representing different variants.

Several insights can be drawn: (i) among the three effect

values, $\gamma$ plays the most critical role, followed by $\beta$. In particular, when both $\beta$ and $\gamma$ are zeroed out, the performance drops drastically. In other cases, performance still degrades but the decline is less severe and relatively consistent. $\gamma$ corresponds to the trajectory of the confidence variations of previously decoded tokens $E_i(p)$ at each decoding step, while $\beta$ represents the trajectory of the one-way influence exerted by the newly decoded token $E_i(n)$ on $E_i(p)$.

The substantial contribution of these two components indicates that DDM's effectiveness relies heavily on modeling both the historical semantic accumulation of tokens (captured by $\gamma$) and the interaction between newly decoded and previously decoded tokens (captured by $\beta$). In other words, DDM goes beyond local confidence signals by leveraging structural and dependancy information from the decoding trajectory, which is key to its superior performance.

**Influence of Decoding Strategy and Token Length.** Besides the influence of effect values, Figure 6 also illustrates the impact of different decoding strategies and token lengths. Focusing on the results of the original DDM (green square), we observe that for CMA (Figures 6c, 6f), these factors have little effect on performance. In contrast, for CCA (Figures 6a, 6b) and IRA (Figures 6d, 6e), performance varies slightly across settings and no consistent pattern emerges, with AUC changes generally within 0.1. Considering that token length and decoding strategies can be flexibly chosen in practice, and that our experiments show their influence on performance to be small, the feasibility of applying DDM in attribution is thus well supported.

**Structure Preservation of GTA.** A primary motivation behind the design of GTA is to better preserve the structural information in DDMs. To assess the necessity and impact of our design, we conduct a comparative study in Figure 7, where the cell-wise Gaussian distribution in GTA

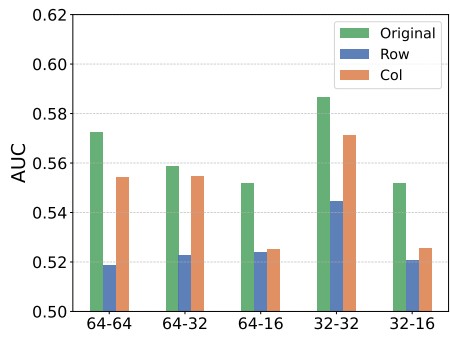

*Figure 7.* Maintaining the structural integrity of DDMs is critical for reliable attribution.

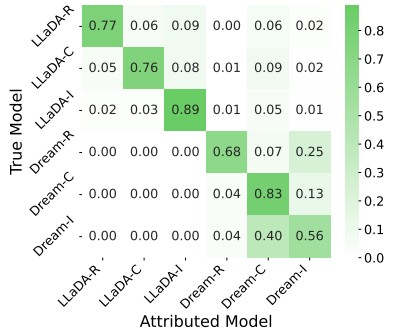

*Figure 8.* Attribution on multiple models within a single attributing process.

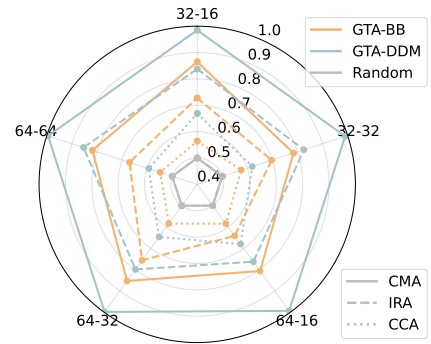

*Figure 9.* The performance of GTA under the black-box setting.

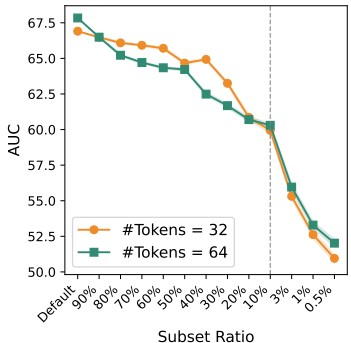

*Figure 10.* Effect of dataset scale for constructing DDM.

*Table 2.* Cross-domain attribution results.

| Info | Setting | AUC | TPR@1%FPR |
|---|---|---|---|
| conf. | In.D | 64.5 | 3.70 |
|  | Crs.D | 59.27 | 1.60 |
| filt. conf. | In.D | 59.53 | 1.87 |
|  | Crs.D | 56.90 | 1.53 |
| **DDM** | In.D | **66.91** | **4.68** |
|  | Crs.D | **62.38** | **3.12** |

*Table 3.* Influence of the effect values.

| std (%) | $\nabla$CMA | $\nabla$IRA | $\nabla$CCA |
|---|---|---|---|
| $\alpha$ | 0.08 | 1.67 | 1.49 |
| $\beta$ | 0.02 | 0.09 | 0.08 |
| $\gamma$ | 0.03 | 0.86 | 0.60 |

is replaced with token-wise (column-based, denoted as Col) and step-wise (row-based, denoted as Row) variants. The results show that when structural information is disrupted, performance drops significantly, particularly in step-wise Gaussian (Row), where the interdependence among tokens is severely undermined, leading to a partial failure of attribution. This further highlights the utility of GTA's design.

**Attribution across Multiple Models.** Although model attribution is typically formulated as a binary classification task, performing attribution across multiple models simultaneously provides a stronger validation of a method's scalability. In Figure 8, we conduct attribution over six model variants used in this work and report the accuracy. We use GSM8K as the dataset in this experiment, with the token length and decoding strategy set to 64 and 32, respectively. Here, *-R* denotes the reference model, *-C* the model compared with *-R* in CCA, and *-I* the model compared with *-R* in IRA. The diagonal entries indicate cases where the predicted model matches the ground-truth model, i.e., attribution is correct. We observe that in most cases, the combination of GTA and DDM consistently achieve reliable attribution, and the limited deviations that arise are predominantly restricted to models within the same family.

**Influence of Effect Values.** Besides the default value of $\alpha$, $\beta$ and $\gamma$ (10, 0.5, 2), we also experiment with other values to demonstrate the robustness of our method to such variations.

As shown in Table 3, we vary the three effect values ($\alpha \in [5, 15]$ step 1; $\beta \in [0.1, 1.0]$ step 0.1; $\gamma \in [1.5, 2.5]$ step 0.1, more variations can be found in Appendix E) and report the AUC std (%) with token length 64 on GSM8K under low-confidence decoding. The std values are mostly below 0.1% and at most around 1%, indicating that DDM and GTA is robust to changes in effect values.

**GTA in Black-box Scenario.** Previous confidence-based methods, including DDM, are operated in the *gray-box* setting, where the adversary lacks access to model parameters or intermediate features and can only rely on model outputs. We further push this boundary by evaluating GTA in the most strict scenario, namely the full *black-box* setting, where the adversary can only observe the final decoded tokens at each step without any confidence information. In this case, GTA constructs distributions directly from the model's decoding history. Results are reported in Figure 9, where GTA-BB refers to GTA under black-box setting. As shown, while the performance degrades, it remains considerable, demonstrating that for dLLMs, the attribution problem is still solvable even under the strictest setting, with their unique decoding nature playing a key role in enabling this.

**Generalizability of GTA.** We evaluate the cross-domain robustness of our method, i.e., how well it performs when the target dataset differs from the one used for model training. In this setting, we use GSM8K for model training and CodeAlpaca-20K is used as the attribution target. For ease of comparison, we denote the default in-domain setting as In.D and the cross-domain setting as Crs.D. Since most other attribution methods fail to operate reliably under the cross-domain scenario, we report only the results of different combinations within GTA. Other dataset configuration and training/attribution pipeline is the same as in Table 1. As can be observed in Table 2, the cross-domain setting is clearly more challenging. While performance does decrease, the drop remains modest, typically within 5%. This indicates that DDM-GTA exhibits strong resilience under cross-domain conditions.

**Influence of data scale for DDM construction.** We con-

duct an experiment on GSM8K to evaluate how the size of the dataset used to construct the DDM affects attribution performance. Specifically, we randomly subsample the training data at 90%, 80%, ..., 10%, 3%, 1%, and 0.5% of the default dataset size and do the DDM construction under each subset. Each sub-sampling scale is repeated five times. We report results for #tokens = 32 and 64, with the block size fixed to 16. As can be observed in Figure 10, when using 10% or more of the data, the AUC consistently remains above approximately 60%. Only in the most extreme setting, i.e., using merely 0.5% of the data, does the performance degrade to the level of random guessing. These results indicate that DDM-GTA has modest data requirements and remains effective even with substantially reduced data.

## 5. Conclusion

In this work, we take a first step toward model attribution for dLLMs. Leveraging the decoding trajectories of dLLMs, we propose the Directed Decoding Map (DDM) to capture interdependencies across decoding steps and reveal model-specific behaviors. Furthermore, We introduce Gaussian Trajectory Attribution (GTA), which models DDMs with cell-wise Gaussian distributions to preserve fine-grained structural information and yield compact probabilistic fingerprints for each model. Extensive experiments demonstrate the effectiveness of our method.

## Acknowledgement

This project is supported by the National Research Foundation, Singapore, and Cyber Security Agency of Singapore under its National Cybersecurity R&D Programme and CyberSG R&D Cyber Research Programme Office (Award: CRPO-GC1-NTU-002).

## Impact Statement

This paper aims to advance machine learning by improving model attribution for dLLMs via decoding-trajectory analysis. Potential positive impacts include better provenance tracking, auditing, and misuse mitigation. As with any attribution method, errors or adversarial evasion may lead to incorrect attribution or overreliance on scores; thus, results should be treated as probabilistic evidence and used with appropriate safeguards and complementary signals.

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

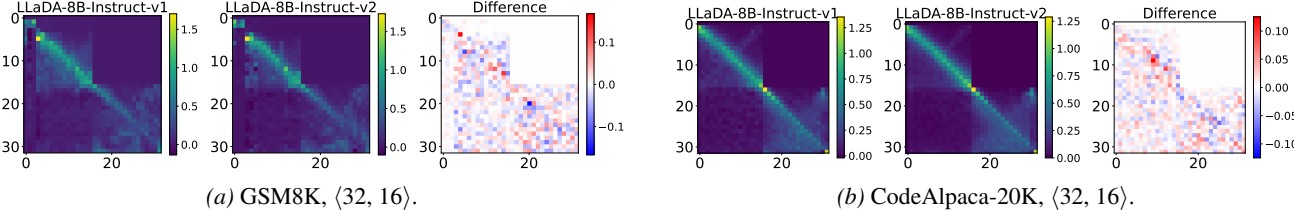

*(a)* GSM8K, $\langle 32, 16 \rangle$.        *(b)* CodeAlpaca-20K, $\langle 32, 16 \rangle$.

*Figure 11.* DDM comparison of LLaDA-8B-Instruct instruction-tuned under two different datasets. $\langle \text{\#tokens, block size} \rangle$ is set to $\langle 32, 16 \rangle$.

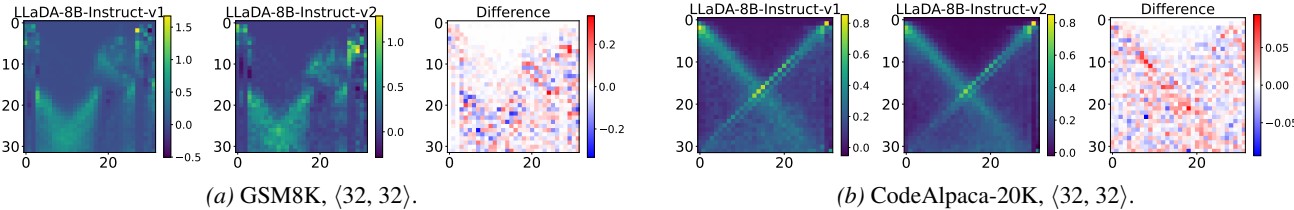

*(a)* GSM8K, $\langle 32, 32 \rangle$.        *(b)* CodeAlpaca-20K, $\langle 32, 32 \rangle$.

*Figure 12.* DDM comparison of LLaDA-8B-Instruct instruction-tuned under two different datasets. $\langle \text{\#tokens, block size} \rangle$ is set to $\langle 32, 32 \rangle$.

## A. More DDM visualizations.

In Figures 11 12 13 14, we further provide several visualizations of DDMs under IRA setting, with the models to be attributed set to LLaDA-8B-Intruct and datasets used are GSM8K and CodeAlpaca-20K. All the five $\langle \text{\#tokens, block size} \rangle$ settings are provided.

## B. More results of Multiple Model Attribution

In Figure 18, we further provide several results under multiple models attribution. The dataset used here is GSM8K and $\langle \text{\#tokens, block size} \rangle$ is set to $\langle 32, 16 \rangle$, $\langle 32, 32 \rangle$, $\langle 64, 16 \rangle$, $\langle 64, 64 \rangle$, respectively.

## C. Another SVD Anylasis Under Semi-supervised Decoding.

In the main text, we analyzed the structural information of DDM using SVD under low-confidence decoding ($\langle \text{\#tokens, block size} \rangle = \langle 32, 32 \rangle$). Here, we provide another example under semi-supervised decoding with $\langle \text{\#tokens, block size} \rangle = \langle 32, 16 \rangle$, as shown in Figure 19. The same phenomenon is observed: different models share almost identical spectra in the leading components, reflecting similar task-level, model-agnostic structures, while clear discrepancies appear in the middle and tail components, which capture low-energy yet highly discriminative directions. This suggests that *attribution signals lie in fine-grained structural patterns rather than dominant modes*, and thus dimensionality reduction or noisy methods that retain only leading components would discard critical differences and weaken attribution.

## D. Training Configurations and Prompts used in Our Work

In Figures 15 and 16, we provide prompt examples for the two datasets used in our work (GSM8K and CodeAlpaca-20K), where GSM8K is for mathematical reasoning task and CodeAlpaca-20K is for code generation task. In Table 4, we report the detailed training configurations of the models used in our paper.

## E. More experiments on the effect values.

Here, we conduct a deeper investigation into the effect values. To eliminate scale-*related biases*, we sample three distinct values from a same range level to serve as the effect values, and three different range levels are used: (0,10.0], (0,100.0], and (0,1000.0]. For each range level, we perform five random samplings and report the averaged results. We set the token length and block size to 32 and 16, respectively. The results under GSM8K and CodeAlpaca-20K are summarized in Table 6. It can be observed that the choice of the effect value has only a negligible impact on performance. Conceptually, the effect value simply serves as a positional marker for different signal types. Nevertheless, some patterns can still be observed from the results. For example, random sampling of the effect value remains effective, but restricting the sampling range to a smaller

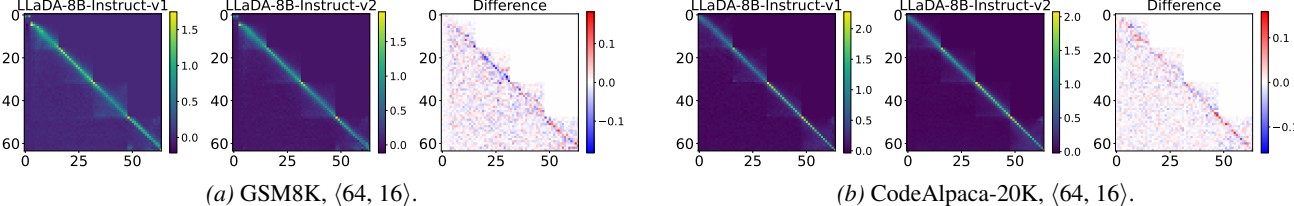

*(a)* GSM8K, $\langle 64, 16 \rangle$.      *(b)* CodeAlpaca-20K, $\langle 64, 16 \rangle$.

*Figure 13.* DDM comparison of LLaDA-8B-Instruct instruction-tuned under two different datasets. $\langle$#tokens, block size$\rangle$ is set to $\langle 64, 16 \rangle$.

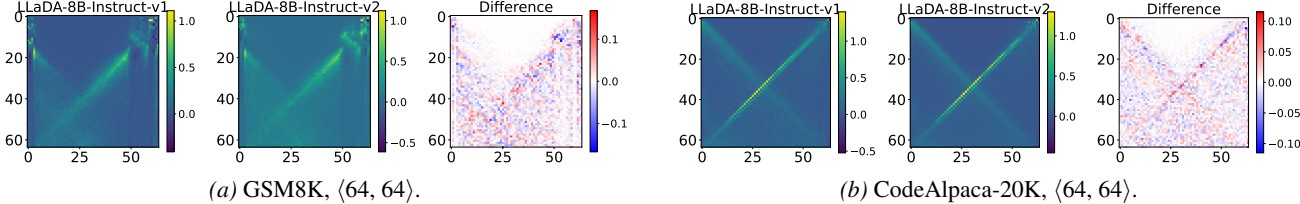

*(a)* GSM8K, $\langle 64, 64 \rangle$.      *(b)* CodeAlpaca-20K, $\langle 64, 64 \rangle$.

*Figure 14.* DDM comparison of LLaDA-8B-Instruct instruction-tuned under two different datasets. $\langle$#tokens, block size$\rangle$ is set to $\langle 64, 64 \rangle$.

*Table 5.* Key symbols and notations.

| Symbol | Description |
|---|---|
| $T$ | Number of decoding steps |
| $L$ | Length of the generated token sequence |
| $c_i(j)$ | Model confidence of position $j$ at decoding step $i$ |
| $\Delta c(j)$ | Confidence change between steps $i$ and $i+1$ at position $j$ |
| $U_i$ | Set of positions already decoded at step $i$ |
| $N_{i+1}$ | Set of positions newly decoded at step $i+1$ |
| $E_i(j)$ | Value of the Directed Decoding Map (DDM) at step $i$ and position $j$ |
| $E_i(n)$ | DDM value for a newly decoded position at step $i$ |
| $E_i(p)$ | DDM value for a previously decoded position at step $i$ |
| $E_i(o)$ | DDM value for a still-masked position at step $i$ (set to 0) |
| $\alpha, \beta, \gamma$ | Positive effect values ($\alpha, \beta, \gamma \in \mathbb{R}_+$) |
| $E$ | DDM matrix, $E \in \mathbb{R}^{T \times L}$ |
| $M$ | A candidate dLLM in the attribution task |
| $K$ | Number of candidate models considered in the attribution task |
| $\mu_M(t, l)$ | Gaussian mean of cell $(t, l)$ in the DDMs of model $M$ |
| $\sigma_M^2(t, l)$ | Gaussian variance of cell $(t, l)$ in the DDMs of model $M$ |
| $\ell_M(E^*)$ | Log-likelihood of a target DDM $E^*$ under model $M$ in GTA |
| $\widehat{M}(E^*)$ | Attributed model for trajectory $E^*$ |

*Table 6.* Attribution accuracy under different DDM ranges.

| | GSM8K (Cobbe et al., 2021) | | | | CodeAlpaca-20K (Chaudhary, 2023) | | | |
|---|---|---|---|---|---|---|---|---|
| | Default | (0, 10.0] | (0, 100.0] | (0, 1000.0] | Default | (0, 10.0] | (0, 100.0] | (0, 1000.0] |
| CMA | 99.95 | $99.87 \pm \mathbf{0.06}$ | $99.79 \pm \mathbf{0.10}$ | $99.65 \pm \mathbf{0.19}$ | 98.94 | $98.84 \pm \mathbf{0.08}$ | $98.76 \pm \mathbf{0.13}$ | $98.62 \pm \mathbf{0.18}$ |
| IRA | 81.75 | $81.84 \pm \mathbf{0.16}$ | $81.59 \pm \mathbf{0.19}$ | $81.37 \pm \mathbf{0.26}$ | 65.05 | $64.96 \pm \mathbf{0.12}$ | $64.79 \pm \mathbf{0.18}$ | $64.63 \pm \mathbf{0.23}$ |
| CCA | 66.91 | $66.59 \pm \mathbf{0.24}$ | $66.44 \pm \mathbf{0.27}$ | $66.40 \pm \mathbf{0.30}$ | 62.64 | $62.55 \pm \mathbf{0.19}$ | $62.47 \pm \mathbf{0.26}$ | $62.32 \pm \mathbf{0.29}$ |

interval leads to more stable outcomes and yields slight (though marginal) performance improvements.

## F. Symbols and Notations.

The key symbols and notations used in our work are given in Table 5.

| Setting | Value |
|---|---|
| *Training Arguments* | |
| GPU | $8 \times$ A6000 |
| Epochs | 20 |
| Batch size | 1 |
| Gradient accumulation | 4 |
| Logging steps | 2 |
| Max seq. length | 4096 |
| Save steps | 100 |
| Learning rate | 1e-5 |
| Weight decay | 0.1 |
| Max grad norm | 1.0 |
| *Deepspeed Config* | |
| Zero stage | 2 |
| Gradient accumulation | 4 |
| Gradient clipping | 1.0 |
| Zero3 init flag | False |
| Processes | 8 |

*Table 4.* Training configuration.

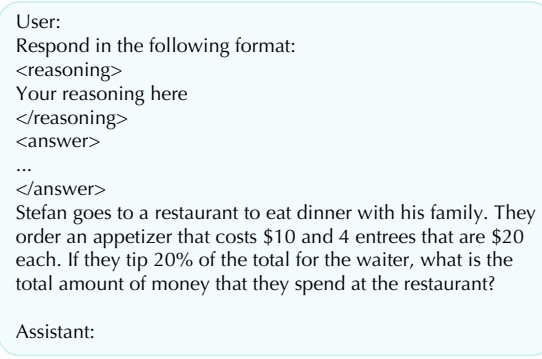

User:
Respond in the following format:
<reasoning>
Your reasoning here
</reasoning>
<answer>
...
</answer>
Stefan goes to a restaurant to eat dinner with his family. They order an appetizer that costs $10 and 4 entrees that are $20 each. If they tip 20% of the total for the waiter, what is the total amount of money that they spend at the restaurant?

Assistant:

*Figure 15.* Prompt for GSM8K.

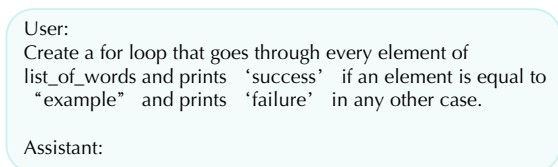

User:
Create a for loop that goes through every element of list_of_words and prints 'success' if an element is equal to "example" and prints 'failure' in any other case.

Assistant:

*Figure 16.* Prompt for CodeAlpaca-20K.

*Figure 17.* Training configuration (left) and prompt examples (right).

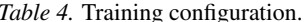
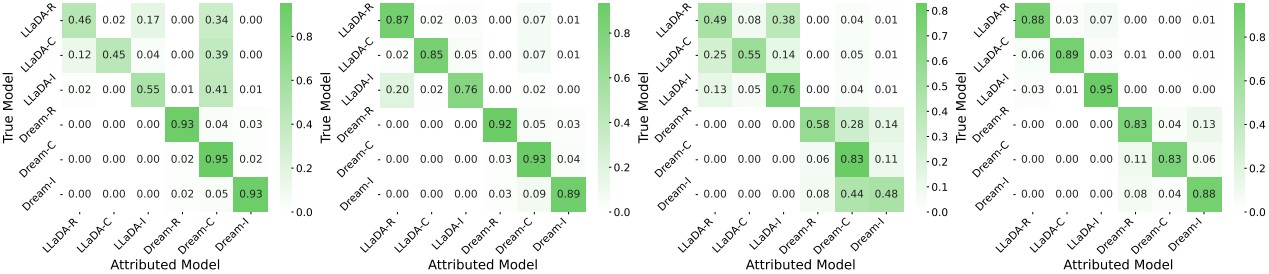

*Figure 18.* Attribution for multiple models under GSM8K. $\langle$#tokens, block size$\rangle$ is set to $\langle 32, 16 \rangle$, $\langle 32, 32 \rangle$, $\langle 64, 16 \rangle$, $\langle 64, 64 \rangle$, respectively.

*Table 7.* Runtime comparison of different attribution methods.

| GSM8K (Cobbe et al., 2021) | | | CodeAlpaca-20K (Chaudhary, 2023) | | |
|---|---|---|---|---|---|
| #Tokens | Method | Time (s) | #Tokens | Method | Time (s) |
| **32** | **GTA (ours)** | 0.38 | **64** | **GTA (ours)** | 0.85 |
| | **Clustering** | 0.49 ($\sim$1.3$\times$) | | **Clustering** | 1.30 ($\sim$1.5$\times$) |
| | **Distance** | 0.73 ($\sim$2$\times$) | | **Distance** | 1.22 ($\sim$1.5$\times$) |
| | **SVD** | 4.53 ($\sim$12$\times$) | | **SVD** | 25.18 ($\sim$30$\times$) |

## G. Efficiency of GTA.

In Table 7, we report the attribution time on the GSM8K target set for all methods. We also include the time required to compute the SVD in this table. We can observe that GTA requires the shortest computation time for attribution and scales well as the number of tokens increases.

## H. Influence of epoch interval.

In our CCA experiment, one model is fully trained for 20 epochs, while the other is the checkpoint saved halfway through training (at 10 epochs). Here, we conduct an additional experiment where the second model is taken from later checkpoints (at 11, 12, ..., 19 epochs). The token length is set to 32 and the block size is 16. The results under LLaDA are shown in

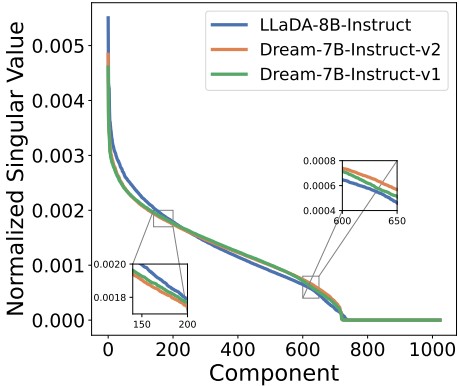

*Figure 19.* A SVD analysis on the structural information of DDMs.

*Table 8.* Attribution accuracy under different epoch intervals (for CCA) on GSM8K.

| | | GSM8K | | | | | | | | | |
|---|---|---|---|---|---|---|---|---|---|---|---|
| Method | Information | 10 (Default) | 9 | 8 | 7 | 6 | 5 | 4 | 3 | 2 | 1 |
| **Distance** | confidence | 61.79 | 59.47 | 58.38 | 56.21 | 53.63 | 52.19 | 50.38 | 50.22 | 50.47 | 50.06 |
| | filtered confidence | 57.34 | 56.83 | 55.12 | 54.47 | 53.26 | 53.05 | 51.42 | 51.01 | 50.72 | 50.06 |
| | **DDM** | **65.84** | **63.48** | **61.69** | **60.06** | **58.59** | **56.40** | **54.37** | **53.69** | **53.24** | **51.28** |
| **GTA** | confidence | 64.50 | 63.32 | 61.03 | 59.80 | 57.12 | 54.26 | 52.80 | 52.18 | 51.36 | 50.62 |
| | filtered confidence | 59.53 | 57.44 | 55.63 | 54.38 | 53.49 | 52.64 | 51.47 | 51.07 | 50.98 | 50.54 |
| | **DDM (ours)** | **66.91** | **64.95** | **62.37** | **61.88** | **60.51** | **58.35** | **57.02** | **55.63** | **53.26** | **52.79** |

Table 8.

Since perplexity-based and clustering-based approaches already perform poorly under our default setting, we focus on the remaining methods. We can observe from the above table that, as the two models become closer in terms of training stage, all methods exhibit a similar trend of gradually decreasing attribution performance, which is expected given the diminishing divergence between the models. However, the DDM–GTA combination consistently performs the best, maintaining an AUC above 55% even when the interval drops to 3 epochs. Among the other methods, the strongest is the DDM–distance combination, which achieves above 55% AUC down to an interval of 5 epochs.

## I. Results under different model intervals.

In our CCA experiment, one model is fully trained for 20 epochs, while the other is the checkpoint saved halfway through training (at 10 epochs). Here, we conduct an additional experiment where the second model is taken from later checkpoints (at 11, 12, ..., 19 epochs). The token length is set to 32 and the block size is 16. The results under LLaDA are shown in Table 8.

## J. Scale and Shift Invariance of Impact Values

This section provides a theoretical justification for why the exact numerical magnitudes of the impact values are not fundamental to the attribution result. Unless otherwise specified, we follow the notation introduced in the main text.

Consider a global affine transformation of the DDM values:

$$E' = aE + b, \qquad a > 0. \tag{7}$$

After refitting the Gaussian statistics under this transformed representation, the mean and variance of each model-specific distribution become

$$\mu'_M = a\mu_M + b, \qquad \sigma'^2_M = a^2\sigma^2_M. \tag{8}$$

*Table 9.* Robustness of GTA under adversarial manipulations on GSM8K with LLaDA under the CCA setup.

| Method | Information | Original | | Pre-hoc steering | | Post-hoc perturb. | |
|---|---|---|---|---|---|---|---|
| | | AUC | TPR@5%FPR | AUC | TPR@5%FPR | AUC | TPR@5%FPR |
| GTA | Confidence | 64.50 | 13.60 | 57.81 | 9.28 | 59.26 | 10.35 |
| GTA | DDM | **66.91** | **14.50** | **63.89** | **11.35** | **64.28** | **12.89** |

For a target value transformed in the same way, i.e.,

$$E'^* = aE^* + b, \tag{9}$$

the normalized quadratic term is invariant:

$$\frac{(E'^* - \mu'_M)^2}{\sigma'^2_M} = \frac{(E^* - \mu_M)^2}{\sigma^2_M}. \tag{10}$$

Therefore, the data-fitting component of the Gaussian log-likelihood remains unchanged under such a transformation.

The normalization term changes only by a constant offset:

$$\log(2\pi\sigma'^2_M) = \log(2\pi\sigma^2_M) + 2\log a. \tag{11}$$

Summing over all $T \times L$ cells yields

$$\ell_M(E'^*) = \ell_M(E^*) - TL\log a. \tag{12}$$

The additional term $-TL\log a$ is independent of the candidate model $M$. Hence, it does not affect the maximizer:

$$\arg\max_M \ell_M(E'^*) = \arg\max_M \ell_M(E^*). \tag{13}$$

This shows that the attribution result is invariant to any global positive scaling and shift of the DDM values. Consequently, the role of the impact values is to provide distinguishable encodings of different DDM states, rather than to impose a uniquely determined numerical scale. The exact magnitudes of these values are therefore not essential, as long as their relative encoding structure is preserved.

## K. Robustness of GTA under Adversarial Manipulations

We further evaluate the robustness of GTA under two adversarial strategies that aim to manipulate the decoding trajectory.

**Pre-hoc steering via in-context learning.** In this setting, we first generate a reference answer using another model and insert it as an in-context example before the target model produces its response. This strategy tests whether the attribution signal can be redirected by conditioning the target model on an externally generated trajectory pattern.

**Post-hoc perturbation during decoding.** We also consider a direct perturbation applied to the decoding logits. Let $z_1$ and $z_2$ denote the top-1 and top-2 logits of each token at a decoding step. We apply a random perturbation:

$$z'_1 = z_1 - \epsilon, \qquad z'_2 = z_2 + \epsilon, \qquad \epsilon \in \{10^{-2}, 5 \times 10^{-2}\}. \tag{14}$$

All other experimental settings follow Table 1. The results on GSM8K with LLaDA under the CCA setup are reported in Table 9. The results show that both adversarial strategies have limited effectiveness in inducing misattribution. Although the performance decreases under manipulation, GTA with DDM information remains consistently stronger than the confidence-based variant. This suggests that the perturbations do not substantially alter the model's overall decoding structure captured by DDM. More effective attacks would likely require modifying the model's decoding dynamics while simultaneously preserving its original task performance, which may require a more carefully designed training-time intervention.

