# OpenReview forum: "Every Step Counts: Decoding Trajectories as Authorship Fingerprints of dLLMs"
_ICML.cc/2026/Conference — ICML 2026 regular_

### Official Review · Reviewer_oGz8 · 2026-03-05

**Soundness:** 2
**Presentation:** 3
**Significance:** 3
**Originality:** 3
**Overall Recommendation:** 4
**Confidence:** 4

**Summary:**

This paper investigates the problem of Model Attribution for discrete diffusion large language models (dLLMs), proposing the use of their unique non-autoregressive decoding trajectories as "fingerprints" for source identification.

Due to the bidirectional nature of dLLMs during decoding, tokens exert mutual influence on one another, causing traditional single-step confidence signals to be highly redundant and obscuring structural information. To address this, the authors design the Directed Decoding Map (DDM) to extract second-order structural features. This matrix precisely records the positive, negative, or mixed impacts—as well as cross-step dependencies—that newly decoded tokens exert on the confidence of previously decoded tokens.

To fully exploit this fine-grained structural information, the paper further proposes Gaussian-Trajectory Attribution (GTA). This method constructs compact probabilistic fingerprints by fitting cell-wise Gaussian distributions to the DDM of each model and performs attribution of target responses using log-likelihood differences. Extensive experiments demonstrate that this approach exhibits strong reliability and robustness in highly challenging attribution scenarios, including cross-model identification, independent fine-tuning from the same checkpoint (maintaining an AUC above 81%), and across different training stages.

**Compliance With Llm Reviewing Policy:**

Affirmed.

**Key Questions For Authors:**

1. Robustness Under Adversarial Decoding StrategiesThe paper validates the effectiveness of DDM and GTA under fixed decoding strategies (such as low-confidence and semi-supervised decoding). However, if an attacker deliberately and dynamically alters the decoding strategy to evade attribution—for instance, by changing block sizes, injecting mask noise, or adjusting sampling temperatures—would the method remain effective?
Scoring Impact: Supplementing the paper with robustness experiments under adversarial decoding strategies, or providing a compelling technical response, would significantly enhance the evaluation of the method's practical security, thereby improving the Soundness score.
2. Theoretical Justification for DDM HyperparametersThe impact values $\alpha, \beta, \gamma$ for the Directed Decoding Map (DDM) are hard-coded to 10, 0.5, and 2, respectively. While Appendix E demonstrates their empirical robustness, the paper lacks a rigorous theoretical explanation for these specific values. Are they merely empirical "placeholders"?
Scoring Impact: Providing a theoretical foundation for these parameter settings, or mathematically proving that the method is invariant to strict monotonic transformations of these states, would substantially strengthen the mathematical rigor of the paper and improve both Originality and Soundness scores.

**Limitations:**

Limited Discriminability for Highly Similar Models: In Cross-Checkpoint Attribution (CCA) experiments, when the training stages of two compared models are extremely close (e.g., an interval of only 1 to 2 epochs), the attribution accuracy drops sharply toward random guessing levels due to the negligible divergence between the models.

Performance Degradation in Black-box and Cross-domain Scenarios: In pure black-box scenarios where generation confidence scores are entirely inaccessible, or in cross-domain settings where the target data differs from the data used to construct the fingerprints, the attribution performance experiences a measurable degree of attenuation.

Heavy Reliance on Prior Data for Fingerprint Construction: The GTA method is not a strictly zero-shot detection approach. Before performing attribution, it must rely on a local dataset to query all candidate target models to extract DDMs and fit their respective Gaussian probability distribution fingerprints.

Vulnerability to Potential Adversarial Attacks: The paper explicitly notes that, similar to other attribution methods, adversarial evasion may cause the method to fail or yield incorrect attribution results. Consequently, it should be deployed in conjunction with other security mechanisms.

**Strengths And Weaknesses:**

Strengths of the Paper
1.Pioneering Contribution: This research marks the first exploration of the model attribution problem within the context of a completely new architecture of large language models (dLLMs).

2.Innovative Methodology Addressing Core Pain Points: The authors astutely identified that the bidirectional decoding nature of dLLMs leads to redundancy in first-order confidence signals. They innovatively proposed the Directed Decoding Map (DDM) to extract second-order structures and cross-step dependencies.

3.Lightweight and Computationally Efficient: Compared to classifier-based methods that incur high training costs, Gaussian-Trajectory Attribution (GTA) is a lightweight statistical feature attribution method. In runtime comparisons, GTA is faster and more scalable than clustering methods or SVD-based decomposition.

4.Robustness in Extreme Scenarios: The method performs exceptionally well under three highly challenging settings: Cross-Model Attribution (CMA), Independent-Run Attribution (IRA), and Cross-Checkpoint Attribution (CCA). Notably, even when two models share the same parameter configuration and are fine-tuned from the same checkpoint, the attribution AUC remains above 81%.

5.Superior Generalization Ability: Experiments show the method remains effective even when using only 10% of the data for construction. Furthermore, it demonstrates attribution resilience in cross-domain datasets and pure black-box scenarios where confidence scores are entirely unavailable.

Weaknesses of the Paper
1.Performance Decay under Extreme Similarity: In the most difficult Cross-Checkpoint Attribution (CCA) experiments, when the training progress of two compared models is very close (e.g., an interval of only 1 or 2 epochs), the attribution AUC drops significantly toward random levels due to the negligible model divergence.

2.Performance Trade-offs in Black-box and Cross-domain Conditions: Although the method remains functional in black-box and cross-domain tests, there is a measurable decrease in performance metrics (such as AUC) compared to gray-box and in-domain scenarios.

3.High Dependency on Structural Information Integrity: GTA relies on cell-wise fitting to preserve trajectory details. If dimensionality reduction techniques (like SVD principal components) or coarse-grained row/column aggregation are forcibly applied, the fine-grained discriminative features are severely compromised, leading to a sharp decline in performance.

4.Data Dependency for Pre-built Fingerprints: Before implementing attribution, GTA requires querying the candidate target models using a local dataset to collect sufficient DDM samples to pre-fit the Gaussian distribution probabilistic fingerprints for each model.

---

> ### Author Rebuttal · Authors · 2026-03-30
>
> It’s our great honor to receive Reviewer oGz8's thoughtful comments and kind words to our work. We would like to clarify several misunderstandings and address the concerns below.
>
> ---
>
> > **[W1,L1]** Performance Decay under Extreme Similarity.
>
> **A1:** Thanks for the comment. We want to emphasize that there is **no practical significance** in distinguishing models that differ by only a single training step; this setting mainly serves as an **extreme stress test**. The difficulty here reflects an inherent limit of the model rather than a limitation of our method. That said, even in this setting, **GTA still consistently outperforms baselines**. We will clarify this in the revision.
>
> ---
>
> > **[W2,L2]** There is a measurable degree in black-box and cross-domain Scenarios.
>
> **A2:** Thanks for the comment. In **black-box settings**, most baselines perform **close to random**, while our method drops by only **~0.1 AUC on average**. In **cross-domain scenarios**, the drop is **~0.05 AUC**, and our method still **significantly outperforms all baselines**. Therefore, **we do not consider this a meaningful degradation**. We will clarify this in the revision.
>
> ---
>
> > **[W3,L3]** Dependency on Structural Integrity.
>
> **A3:** Thanks for the comment. This is **not a limitation of our method**. There are two things we want to respectfully clarify.
>
> (i) The **structural information** is **not an additional requirement**, but an **intrinsic signal naturally available in dLLMs**. Recent diffusion-LLM APIs already expose intermediate decoding trajectories. For example, **Inception’s Mercury API** provides a *diffusing mode* that explicitly output intermediate decoding steps before the final output, corresponding to the **black-box setting in Fig. 9 (Lines 420–434)**.
>
> (ii) Coarse-grained aggregation (Fig.7) and the SVD experiment (Fig.2) are **exploratory analyses included for completeness**. In practice, once the full decoding trajectory is available, there is **no reason for an adversary to aggregate rows/columns or apply dimensionality reduction**, and existing dLLM decoding processes **do not perform such operations**.
>
> ---
>
> > **[W4,L4]** Data Dependency for Pre-built Fingerprints.
>
> **A4:** Thanks for the comment. We would like to respectfully clarify two points.
>
> (i) As shown in **Fig. 10** (Lines 411–425), attribution performance drops to near random **only when using 0.5% of the dataset**. For GSM8K, this is **about 13 samples** (8.79k × 0.3 × 0.005 ≈ 13), indicating that the data requirement for building fingerprints is **extremely small**.
>
> (ii) **Table 2** evaluates the **cross-domain performance of GTA**. The results **remain strong and consistently outperform baselines**, suggesting that even when the original source data is unavailable, **data from other domains can still construct effective fingerprints**.
>
> ---
>
> > **[Q1]** Robustness under dynamic decoding strategies.
>
> **A5:** Thanks for highlighting this important consideration. Here, we try two different adversarial methods: **(1) Post-hoc perturbation during decoding.** Let $z_1$ and $z_2$ denote the top-1 and top-2 logits of each token in a decoding step. We apply a small perturbation: $z_1' = z_1 - \epsilon,; z_2' = z_2 + \epsilon$ ($\epsilon \in [1e-2,5e-2]$). **(2) Dynamic Decoding with AdaBlock-dLLM [1]**. All other settings follow Table 1. Results on GSM8K with LLaDA under the CCA setup are shown below.
>
> |Original||Post-hoc perturb||AdaBlock||
> |-|-|-|-|-|-|
> AUC|TPR@5%FPR|AUC|TPR@5%FPR|AUC|TPR@5%FPR|
> 66.91|14.50|64.28|12.89|63.17|12.13|
>
> As observed, both attacks have limited success in inducing misattribution. We will include the discussion in the revision.
>
> [1] AdaBlock-dLLM: Semantic-Aware Diffusion LLM Inference via Adaptive Block Size
>
> ---
>
> > **[Q2]** Providing a theoretical foundation for the impact values.
>
> **A4:** Thanks for the valuable suggestion. Here is a simple observations explain why the exact magnitudes are not critical (unless otherwise specified, we adopt the notation from our manuscript).
>
> Let $E' = aE + b, a>0$. After refitting Gaussian statistics, $\mu'_M = a\mu_M + b, \sigma'^2_M = a^2\sigma_M^2$. Thus, for a target $E'^* = aE^* + b$, we have:
>
> $$\frac{\left(E'^* - \mu'_M\right)^2}{\sigma'^2_M} = \frac{\left(E^* - \mu_M\right)^2}{\sigma_M^2}$$
>
> The log term becomes
>
> $\log(2\pi\sigma'^2_M)=\log(2\pi\sigma_M^2)+2\log a$.
>
> Summing over all $T\times L$ cells gives:
>
> $\ell_{M}(E^{\prime \*})=\ell_{M}(E^{\*})-TL\log a$
>
> The offset is independent of $M$, therefore
>
> $\arg\max_M \ell_{M}(E^{\prime \*})=\arg\max_M \ell_M(E^{\*})$.
>
> Thus, the attribution result is invariant to global scaling/shift of DDM values. The role of the impact values is only to provide separated encodings of different DDM states, and the exact magnitudes are not fundamental.
>
> ---
>
> Thanks again for the thoughtful and constructive feedback. **If Reviewer oGz8 still has any remaining concerns after reading our rebuttal, please let us know so that we can clarify further.**

---

> > ### Author Rebuttal · Reviewer_oGz8 · 2026-04-01
> >
> > Thank you for your detailed response. It has fully addressed my concerns, and I have no further questions.

---

> > > ### Author Response · Authors · 2026-04-01
> > >
> > > We are more than encouraged to hear that our response has fully addressed the concerns of Reviewer oGz8, and we are grateful that Reviewer oGz8 increased the score to 4.
> > >
> > > Thanks again to Reviewer oGz8 for the great effort in reviewing our work and for the valuable comments.

---

### Official Review · Reviewer_xFaY · 2026-03-11

**Soundness:** 2
**Presentation:** 3
**Significance:** 2
**Originality:** 3
**Overall Recommendation:** 4
**Confidence:** 4

**Summary:**

This paper is targeting at the model attribution task. Technically, the authors find the decoding trajectory of dLLM is a good signiture to attribute to different dLLMs. Thus, the authors proposed "a information extraction scheme called the Directed Decoding Map (DDM), which captures structural relationships between decoding steps and reveals model-specific behaviors". Based upon this feature, the authors propose Gaussian-Trajectory Attribution (GTA) approach to get the loglikelihood differences as the scoring function for model attribution.
Basically, the findings of using decoding trajectory to attribute dLLM outputs is novel. However, my major concern is that the problem setting is not practical as the real application typically do not provide such decoding trajectory and the setting is usually black-box, all the user has is the response texts. In other words, the motivation of the paper is very weak.

**Compliance With Llm Reviewing Policy:**

Affirmed.

**Final Justification:**

my concerns are addressed.

**Key Questions For Authors:**

Can you provide solid and convincing application scenarios the tehchnique would be used?

**Limitations:**

yes

**Strengths And Weaknesses:**

Pros:

1. the paper is well-writtern and well-structured, and easy to follow.

2. The paper is thoroughly validated, the experiments are convincing.

Cons:

1. As mentioned, my major concern is that the problem setting is not practical as the real application typically do not provide such decoding trajectory and the setting is usually black-box, all the user has is the response texts. In other words, the motivation of the paper is very weak. In real practice, how can you obtain these decoding trajectories? Without convincing motivation, no matter how elegant and solid solution it is, the work is not sound.

2. Many AI detection or model attribution works are not cited, discussed. As a research paper, a thorough related work review is highly encouraged. Here are some examples of AI-detection for references:
Detectgpt: Zero-shot machine-generated text detection using probability curvature， ICML'23.
Radar: Robust ai-text detection via adversarial learning. NeurIPS'23.
Dna-gpt: Divergent n-gram analysis for training-free detection of gpt-generated text. ICLR'24.
Fast-detectgpt: Efficient zero-shot detection of machine-generated text via conditional probability curvature. ICLR'24.
Spotting llms with binoculars: Zero-shot detection of machine-generated text. ICML'24.
Ghostbuster: Detecting Text Ghostwritten by Large Language Models. ACL'24
Dald: Improving logits-based detector without logits from black-box llms. NeurIPS'24.
Detectllm: Leveraging log rank information for zero-shot detection of machine-generated text. NeurIPS'24.
Human Texts Are Outliers: Detecting LLM-generated Texts via Out-of-distribution Detection. NeurIPS'25.

---

> ### Author Rebuttal · Authors · 2026-03-30
>
> It’s our great honor to receive Reviewer xFaY's thoughtful comments and kind words to our work. We would like to address the concerns as below.
>
> ---
>
> > **[C1,Q1]** Real application provides the decoding trajectory.
>
> **A1:** Thanks for highlighting this important consideration. **Emerging commercial diffusion-LLM APIs already expose intermediate decoding trajectories**. For example, **Inception’s Mercury API** (see the official documents from the website of inceptionlabs) provides a **diffusing mode** that explicitly output intermediate decoding steps before getting the final output, which corresponds to our black-box setting in Figure 9, Lines 420-434. Besides, other closed-source models such as **Gemini Diffusion**, although not yet publicly available, show in their **official demos** that they provide both **real-time outputs and slowed-down outputs**, where the latter makes the decoding trajectory **visible to users**. We will **add these examples into our introduction** to further strengthen the motivation.
>
> ---
>
> > **[C2]** Add the mentioned related works.
>
> **A2:** Thanks for the mentioning these valuable works. We will **add a new paragraph** in the **Authorship Attribution** subsection (begin at Line 137) of the Related Works section and thoroughly **discuss the works you mentioned explicitly**. The raw content is as follows:
>
> - Recent studies have also developed a large body of text-only detectors for machine-generated content. **DetectGPT [1]** exploits probability curvature, showing that machine-generated text tends to lie in negative-curvature regions of the model’s log-probability landscape... **RADAR [2]** improves robustness by adversarially co-training a paraphraser and a detector to resist paraphrasing attacks... **DNA-GPT [3]** proposes divergent n-gram analysis by regenerating continuations from truncated prefixes for training-free detection... **Fast-DetectGPT [4]** improves the efficiency of curvature-based zero-shot detection via conditional probability curvature... **Binoculars [5]** contrasts the outputs of two related language models to enable strong zero-shot detection without task-specific training... **Ghostbuster [6]** extracts features from weaker language models and trains a classifier for black-box detection without target logits... **DALD [7]** aligns surrogate-model distributions to enable logits-based detection when the original logits are unavailable... **DetectLLM [8]** leverages token log-rank statistics to provide efficient zero-shot detection signals... **Human Texts Are Outliers [9]** reframes detection as an out-of-distribution problem where human-written text is treated as the outlier class... In contrast to these works, our work focuses on **model attribution for diffusion language models (dLLMs)** by exploiting their decoding trajectories as model-specific fingerprints.
>
> [1] DetectGPT: Zero-Shot Machine-Generated Text Detection Using Probability Curvature
>
> [2] RADAR: Robust AI-Text Detection via Adversarial Learning
>
> [3] DNA-GPT: Divergent N-Gram Analysis for Training-Free Detection of GPT-Generated Text
>
> [4] Fast-DetectGPT: Efficient Zero-Shot Detection of Machine-Generated Text via Conditional Probability Curvature
>
> [5] Spotting LLMs With Binoculars: Zero-Shot Detection of Machine-Generated Text
>
> [6] Ghostbuster: Detecting Text Ghostwritten by Large Language Models
>
> [7] DALD: Improving Logits-Based Detector Without Logits From Black-Box LLMs
>
> [8] DetectLLM: Leveraging Log Rank Information for Zero-Shot Detection of Machine-Generated Text
>
> [9] Human Texts Are Outliers: Detecting LLM-Generated Texts via Out-of-Distribution Detection
>
>
> ---
>
> Thanks again for the thoughtful and constructive feedback. If reviewer xFaY has any remaining concerns, we would definitely love to clarify further.

---

> > ### Author Rebuttal · Reviewer_xFaY · 2026-04-01
> >
> > Thanks for the detailed feedback. My concerns are resolved. I would happy to raise my score to 4.

---

> > > ### Author Response · Authors · 2026-04-01
> > >
> > > We are very grateful to hear that our response has well addressed the concerns of Reviewer xFaY, and we sincerely appreciate Reviewer xFaY's strong support for our work.
> > >
> > > Thanks again to Reviewer xFaY for the thoughtful and constructive feedback throughout the review process.

---

### Official Review · Reviewer_2u6F · 2026-03-13

**Soundness:** 4
**Presentation:** 3
**Significance:** 4
**Originality:** 4
**Overall Recommendation:** 5
**Confidence:** 4

**Summary:**

The authors describe the need to learn attribution to new dLLMs, specifically determining model, model training run, and checkpoint via GTA applied to a DDM (where the authors assume a gray box scenario, in which they see the step-by-step output of the model but not its model weights). GTA likelihood ratios are used to assess similarity or differences between samples  of DDMs. The authors achieve SOTA performance compared to reasonable baselines across 2 dLLM models, 2 training datasets, and various scenarios: model attribution, model training run attribution, and checkpoint attribution.

**Compliance With Llm Reviewing Policy:**

Affirmed.

**Final Justification:**

My concerns are addressed.

**Key Questions For Authors:**

How does GTA compare against distances based on SVD differences from a model’s DDM SVD “template”?

**Limitations:**

Yes

**Strengths And Weaknesses:**

Strengths:
Soundness
The authors show very good results across models and scenarios, showing they achieve SOTA for the gray model scenario
The authors also explore the more realistic black box scenario where the researchers observe tokens decoded at each step. The performance in this scenario is very useful, and indeed should be emphasized earlier. While I had questions about the realism of the gray box scenario, the black box scenario’s ability to capture attribution is very helpful.
Ablation, meanwhile conclusively demonstrates their performance is due to DDM’s ability to capture accumulation of tokens and interactions between newly decoded and previously decoded tokens.
Presentation
Overall, the methods were clear. I was very impressed. That said the overall emphasis on gray-box results leaves users to question the utility of the method as we should expect black box scenarios more likely. I am glad it was mentioned albeit a little too late.
Significance
The work appears significant, as I think attribution is an important point.
Originality
Attribution to dLLMs seems novel. I have no qualms with novelty.

Weaknesses
Soundness
My biggest concern is that while the method can distinguish between two LLMs, it is unclear if it can distinguish between many other LLMs, or if performance drops significantly
The authors mention prior ways that LLMs were able to be distinguished from humans, e.g., GPTZero or DetectGPT. Following that analogy, note that somewhat unethical start-ups like StealthGPT, BypassGPT, and HIX Bypas can make LLM detection more difficult. Therefore, in the scenario of LLM attribution, how brittle is GTA to adversarial methods? Is someone wanted to obfuscate the dLLM to output results that look like another dLLM, could we detect this? This is somewhat outside of the paper’s scope, but I am not fully convinced that the method could very robustly determine attribution.
More generally, the authors should leave room to discuss limitations.

Presentation
Figure 6 is pretty confusing with the many greek letters. Is there a way to present this to more clearly denote what has changed from each ablation?
Figure 5 shaded areas are not explained
Table 1 parentheticals (“(^30.43)”) seem confusing - why compare against perplexity alone?
Significance
N/A
Originality
N/A

---

> ### Author Rebuttal · Authors · 2026-03-30
>
> It’s our great honor to receive Reviewer 2u6F's thoughtful comments and strong support for our work. We would like to address the concerns as below.
>
> ---
>
> > **[W1]** Whether our method can distinguish beyond two models.
>
> **A1:** Thanks for the valuable comment. In **Fig.8**, we evaluate our method when **attributing 5 models simultaneously** (see Lines 364–379, right side). Here, we further include another two models: **SDAR-30B-A3B-Chat [1]** and **dLLM-Var [2]**. The dataset configuration and training/attribution pipeline follow Table 1. For **SDAR-30B-A3B-Chat**, we adopt the static decoding strategy, while for **dLLM-Var** we use its default decoding setup. Results on GSM8K under the difficult CCA setting are shown below:
>
> |||SDAR-30B-A3B-Chat [1]||dLLM-Var [2]||
> |-|-|-|-|-|-|
> |Method|Information|AUC|TPR@5%FPR|AUC|TPR@5%FPR|
> |Distance|confidence|58.86|4.76|57.45|3.85|
> ||**DDM**|**63.81**|**9.31**|**62.96**|**8.32**|
> |**GTA**|confidence|61.59|15.72|60.73|11.48|
> ||**DDM (ours)**|**65.98**|**17.65**|**67.06**|**15.84**|
>
> As can be observed, DDM with GTA remains the most effective. Also, when DDM is cooperated with other attribution methods, it can still help to improve the performance.
>
> We also conduct another experiment on **attributing between the four models simultaneously** (LLaDA, Dream, SDAR and dLLM-Var). The setting is the same as in Figure 8, and we report the diagonal results on GSM8K below:
>
> |LLaDA|Dream|SDAR-30B-A3B-Chat|dLLM-Var|
> |-|-|-|-|
> |0.83|0.78|0.75|0.80|
>
> We can observe that the attribution is still successful. We will include the above discussion in the revision.
>
> [1] SDAR: A Synergistic Diffusion-AutoRegression Paradigm for Scalable Sequence Generation
>
> [2] Diffusion LLM with Native Variable Generation Lengths: Let [EOS] Lead the Way
>
> ---
>
> > **[W2]** How brittle is GTA to adversarial methods?
>
> **A2:** Thanks for highlighting this important consideration. Here, we try to explore two adversarial strategies to manipulate the decoding trajectory: **(1) Pre-hoc steering via in-context learning.** We first generate a reference answer using another model and insert it as an in-context example before the target model produces its response. **(2) Post-hoc perturbation during decoding.** Let $z_1$ and $z_2$ denote the top-1 and top-2 logits of each token in a decoding step. We apply a random perturbation: $z_1' = z_1 - \epsilon,; z_2' = z_2 + \epsilon$ ($\epsilon \in [1e-2,5e-2]$). All other settings follow Table 1. Results on GSM8K with LLaDA under the CCA setup are shown in the table below.
>
> |||Original||Pre-hoc steering||Post-hoc perturb||
> |-|-|-|-|-|-|-|-|
> |Method|Information|AUC|TPR@5%FPR|AUC|TPR@5%FPR|AUC|TPR@5%FPR|
> |**GTA**|confidence|64.50|13.60|57.81|9.28|59.26|10.35|
> ||**DDM (ours)**|**66.91**|**14.50**|**63.89**|**11.35**|**64.28**|**12.89**|
>
> As observed, both attacks have limited success in inducing misattribution. This is likely because the perturbations do not significantly alter the model’s overall decoding structure. However, achieving this while preserving a model's original performance requires a carefully designed training process. We will **include this discussion in the revision**.
>
> ---
>
> > **[W3]** Some presentation issues.
>
> **A3:** Thanks for pointing this out. We will revise the presentation of the mentioned figures and tables to improve clarity.
>
> Specifically, for **Figure 6**, we will merge the five points of each color into a single point and report the **mean with error bars**, which makes the figure more concise and easier to interpret without affecting the conclusions.
>
> Regarding the **shaded areas in Figure 5**, besides the current brief explanation in Line 310 (right side), we will add a clearer description: under the CMA setting, some baselines (e.g., clustering) show higher variance, likely because models produce highly similar high-dimensional representations that clustering fails to distinguish.
>
> For **Table 1**, we will revise the parenthetical comparisons to highlight differences between **information extraction schemes within each method**, making the comparison clearer and more informative.
>
> ---
>
> > **[Q1]** How does GTA compare against distances based on SVD differences from a model’s DDM SVD template.
>
> **A4:** Thanks for the valuable comment. Follow your suggestion (other settings are the same as in Table 1), we report the results on GSM8K with LLaDA under the CCA setup in the table below:
>
> |Method|Information|AUC|TPR@5%FPR|
> |-|-|-|-|
> |Distance|SVD|61.50|6.90|
> |Distance|DDM|65.84|8.61|
> |GTA (ours)|DDM|66.91|14.50|
>
> The low performance of SVD templates is because they mainly preserve dominant components while discarding fine-grained structural patterns, whereas useful attribution signals lie in the middle and tail of the spectrum (see discussions from Line 208). We will add this discussion in the revision.
>
> ---
>
> Thanks again for the thoughtful and constructive feedback. If reviewer 2u6F has any remaining concerns, we would be happy to clarify further.

---

> > ### Author Rebuttal · Reviewer_2u6F · 2026-03-31
> >
> > The weaknesses mentioned have been addressed. Even discussing as limitations would be enough, but I see the additional analyses were convincing.

---

> > > ### Author Response · Authors · 2026-04-01
> > >
> > > We are more than encouraged to hear that our response has well addressed the concerns of Reviewer 2u6F, and we are **truly greatful for reviewer 2u6F's strong support for our work**.
> > >
> > > Thanks again for reviewer 2u6F's thoughtful and constructive feedback throughout the review process.

---

### Decision · Program_Chairs · 2026-04-30

**Decision:**

Accept (regular)

**Comment:**

This paper presents the first study to utilize the decoding trajectory of Discrete Diffusion LLMs (dLLMs) as a fingerprint for model attribution. Starting from the observation that traditional step-by-step confidence signals become highly redundant due to the bidirectional nature of dLLMs, the authors propose the Directed Decoding Map (DDM) to capture structural relationships between decoding steps, along with Gaussian-Trajectory Attribution (GTA). The framework reports SOTA performance across cross-model, independent-run, and cross-checkpoint attribution in various scenarios, including gray-box and black-box settings.

The paper pioneers the novel problem of dLLM model attribution and demonstrates compelling performance across multiple scenarios through the theoretically grounded DDM+GTA framework. All three reviewers supported an Accept decision with high confidence, and all concerns were resolved during the rebuttal. Accordingly, the AC has decided to Accept the paper.

As the authors committed during the rebuttal to include experiments on simultaneous attribution of five models, adversarial decoding, and SVD comparisons, these must be faithfully incorporated into the camera-ready version.